REGISTERED REPORT PROTOCOL

# Protocol for assessing stakeholder engagement in the development and evaluation of the Informed Health Choices resources teaching secondary school students to think critically about health claims and choices

**Allen Nsangi[1,2], Andrew David Oxman[3]\*, Matt Oxman[3,4], Sarah E. Rosenbaum[3], Daniel Semakula[1,2], Ronald Ssenyonga[1,2], Michael Mugisha[2,5], Faith Chelagat[2,6], Margaret Kaseje[6], Leaticia Nyirazinyoye[5], Iain Chalmers[3], Nelson Kaulukusi Sewankambo[1]**

**1** College of Health Sciences, Makerere University, Kampala, Uganda, **2** Faculty of Medicine, University of Oslo, Oslo, Norway, **3** Centre for Informed Health Choices, Norwegian Institute of Public Health, Oslo, Norway, **4** Faculty of Health Sciences, Oslo Metropolitan University, Oslo, Norway, **5** University of Rwanda, Kigali, Rwanda, **6** Tropical Institute of Community Health and Development (TICH), Kisumu, Kenya

\* oxman@online.no

This is a Registered Report and may have an associated publication; please check the article page on the journal site for any related articles.

## Abstract

### Background

As part of a five year plan (2019–2023), the Informed Health Choices Project, is developing and evaluating resources for helping secondary school students learn to think critically about health claims and choices. We will bring together key stakeholders; such as secondary school teachers and students, our main target for the IHC secondary school resources, school administrators, policy makers, curriculum development specialists and parents, to enable us gain insight about the context.

### Objectives

1. To ensure that stakeholders are effectively and appropriately engaged in the design, evaluation and dissemination of the learning resources.

2. To evaluate the extent to which stakeholders were successfully engaged.

### Methods

Using a multi-stage stratified sampling method, we will identify a representative sample of secondary schools with varied characteristics that might modify the effects of the learning resources such as, the school location (rural, semi-urban or urban), ownership (private, public) and ICT facilities (under resourced, highly resourced). A sample of schools will be randomly selected from the schools in each stratum. We will aim to recruit a diverse sample of

**Data Availability Statement:** The data files
resulting from the stakeholder engagement
processes will be made available at the Norwegian
Centre for Research Data (http://www.nsd.uib.no/
nsd/english/index.html).

**Funding:** This work will be funded by the Research
Council of Norway (https://www.forskningsradet.
no/en/), Project number 284683, grant no:69006
awarded to ADO. The funder will play no role in the
study design, data collection, data analysis, data
interpretation, or writing of the report. The principal
investigator will have access to all the data in the
study and have final responsibility for the decision
to submit for publication.

**Competing interests:** The authors have declared
that no competing interests exist.

**Abbreviations:** IHC, Informed Health Choices
Project.

students and secondary school teachers from those schools. Other stakeholders will be purposively selected to ensure a diverse range of experience and expertise.

## Results

Together with the teacher and student networks and the advisory panels, we will establish measurable success criteria that reflect the objectives of engaging stakeholders at the start of the project and evaluate the extent to which those criteria were met at the end of the project.

## Conclusion

We aim for an increase in research uptake, improve quality and appropriateness of research results, accountability and social justice.

## Background

Previously,the Informed Health Choices (IHC) project, has developed and evaluated resources for helping primary school children learn to think critically about health claims and choices [1]. An example of a common claim in East Africa is, "*Cow dung as a treatment for burns*". Good choices depend on health literacy ie, people's ability to obtain, process, understand and judge the reliability of relevant health information, but often people tend to overestimate treatment benefits and underestimate treatment harms, when making health choices [2].

Moreover, improving health literacy in particular people's ability to assess claims about treatment effects, has the potential to reduce unnecessary suffering and save resources.

For the development of the primary school resources, we brought together various stakeholders and end users, such as children, teachers and policymakers, into the different phases of our work [3]. This was particularly important since none of the researchers belonged to the end user-groups for which we were developing resources. Teachers were included as collaborators through brainstorming [3] and prototyping workshops; policymakers were periodically informed of progress and consulted; and children's feedback on the different versions of the resources was sought early on in the project through workshops and school visits. This enabled the research team to gain insight about the context and stakeholders, formulate ideas, sketches and prototypes.

Building on that work, we are now developing and evaluating resources for secondary school students. We will engage teachers, students, policymakers and other stakeholders throughout this work. The purpose of this protocol is to describe how we will ensure that stakeholders are effectively and appropriately engaged in the design, evaluation, and dissemination of the learning resources; and how we will evaluate the extent to which stakeholders are successfully engaged.

As with the previous project, stakeholders were not involved in the project grant application, project design or in the development of this or other protocols.

In this protocol, we will use the following definition of 'Stakeholder engagement' and 'stakeholders' [4]:

**Stakeholder engagement** is "an iterative process of actively soliciting the knowledge, experience, judgment and values of individuals selected to represent a broad range of direct interests in a particular issue, for the dual purposes of:

- creating a shared understanding

- making relevant, transparent and effective decisions"

**Stakeholders** are "individuals, organisations or communities that have a vested interest in the process and outcomes of the project, research or policy endeavour."

Whilst there has been growing interest and demand for stakeholder engagement in health research [5–13], there has also been growing interest in stakeholder or user engagement in educational research [13]. The terms 'stakeholder engagement' (commonly used in North America) and 'patient and public involvement' (PPI) in research (commonly used in the UK) are sometimes used interchangeably [14]. However, there are important differences in the terminology that is used. For example, INVOLVE (in the UK) defines 'public involvement' in research as research being carried out 'with' or 'by' members of the public rather than 'to', 'about' or 'for' them [15], and uses the following definitions:

- **Involvement**–where members of the public are actively involved in research projects and in research organisations

- **Participation**–where people take part in a research study

- **Engagement**–where information and knowledge about research is provided and disseminated

The International Association for Public Participation (IAP2) [16] seeks to promote and improve the practice of public participation or community and stakeholder engagement more broadly–not specifically in research. In contrast to the definitions used by INVOLVE, it uses the following definitions to describe different degrees of participation in decision making:

- **Inform**–To provide the public with balanced and objective information to assist them in understanding the problem, alternatives, opportunities and/or solutions

- **Consult**–To obtain public feedback on analysis, alternatives and/or decisions

- **Involve**–To work directly with the public throughout the process to ensure that public concerns and aspirations are consistently understood and considered

- **Collaborate**–To partner with the public in each aspect of the decision including the development of alternatives and the identification of the preferred solution

- **Empower**–To place final decision making in the hands of the public.

In this protocol, we will use terminology derived from the International Association for Public Participation (IAP2) and Bruns' extended ladder of participation [17, 18].

Our aim is to increase the level of public impact by informing, consulting, involving and collaborating with others in the decision making process while retaining some authority in the final decisions. In Brun's extended ladder of participation [17], this process is deliberately arranged horizontally (as seen in Table 1) to suggest a range of options rather than a hierarchy.

Stakeholders can be engaged to various degrees in designing, evaluating, and disseminating learning resources. The objectives of different levels of stakeholder engagement, what stakeholders can expect at each level, and ways of achieving different levels of engagement are summarised in (Table 1).

## Potential benefits and harms of stakeholder engagement

Hypothesised benefits of engaging stakeholders in research include better quality and appropriateness of research, empowering people, increasing uptake of results, accountability, and

**Table 1. Levels of stakeholder engagement in designing, evaluating, and disseminating learning resources***.

|  | Information | Consultation | Involvement | Collaboration | Delegation |
|---|---|---|---|---|---|
| **Objectives of stakeholder engagement** | To provide stakeholders with information in order to help them understand the need for the resources | To obtain specific types of input from stakeholders, such as feedback on the resources (including their usefulness and value) or input into the design, evaluation, or dissemination of the resources | To work directly with stakeholders in designing the resources, to ensure that their views are understood and considered, or to engage them in deliberations about problems and proposed solutions | To partner with stakeholders throughout the process of designing, evaluating, and disseminating the resources | To give control over aspects of the design, evaluation, and dissemination of the resources |
| **What stakeholders can expect** | To be kept informed | To be listened to, and provided with information about how their input has influenced subsequent decisions | To work together in designing, evaluating, and disseminating the resources | To jointly work through problems and proposed solutions that will be incorporated as far as possible. | To deliberate on final decisions and participate in consensus processes. |
| **Ways of achieving the objective** | One-way information dissemination such as: • A website • Tailored information (simplified and translated in various languages if required) • Presentations | Two-way communication which involves seeking input, listening, and the exchange of views. This may take the form of: • Written comments • Interviews • Focus groups • Surveys | Interactive discussion and dialogue: • Workshops • Working groups | Stakeholder representatives "at the table", and active as team members in the design, evaluation, and dissemination of the resources. Stakeholders are not involved in final decisions but are involved in: • Advisory groups • Consensus processes | Decisions by a group or organisation with specific authorisation: • Delegation of authority to participate in the decision-making processes about the design, evaluation, or dissemination of the resources |

* Adapted from IAP2 guidance and Bruns extended ladder of participation [16, 17, 19].

social justice [6, 11]. Other, more specific potential benefits include user-relevant research questions; user-friendly information, questionnaires, and interview schedules; more appropriate recruitment strategies; user-focused interpretation of data; and improved dissemination of study results [9]. However, few studies have formally evaluated the extent to which stakeholder engagement has achieved these benefits and very few have compared alternative ways of engaging stakeholders [5–14].

In addition to concerns about a lack of evidence about the benefits of stakeholder engagement, several other concerns have been raised. Poorly planned and implemented efforts to engage stakeholders can create mistrust, waste people's time, and undermine future attempts at engagement [18]. Other concerns include [8, 9, 13, 20]:

- the risk of tokenism and superficiality;

- a lack of clarity about the nature and purpose of stakeholder engagement;

- researchers not having the necessary skills, time and resources to effectively manage stakeholder engagement processes;

- individual stakeholders dominating decisions (unequal participation and unbalanced representation);

- the boundaries between stakeholders and researchers becoming blurred over time, resulting in stakeholders losing their perspective as users of the results;

- potential conflicts between the perspectives of researchers and stakeholders regarding criteria of quality and appropriateness; and

- slowing down research, making it costlier, and opportunity costs (forgone opportunities as a result of using resources on stakeholder engagement).

As with the hypothesised benefits, there is very little evidence of the extent to which stakeholder engagement has had undesirable effects or the costs of stakeholder engagement processes [5–13].

Our purposes for engaging stakeholders in various aspects of designing, evaluating and disseminating IHC learning resources for secondary schools are to ensure that:

- Any concerns they have are heard and considered

- Problems are analysed, described and perceived correctly

- Appropriate solutions are identified

- Important barriers to scaling up use of the learning resources are identified and addressed

- Promising implementation strategies are identified

- Decisions arising from collected feedback are appropriate and acceptable

- The results of the research are meaningful and accessible to them

However, efforts to engage stakeholders should respect the time they have available and the value of their potential contributions. In addition to having a clear purpose, stakeholders' input should be considered, and it should be made clear to them when and how they are able to influence decisions. To ensure that stakeholders are appropriately engaged in the development of the IHC secondary school resources, we will systematically consider how we can best engage them in the different phases of the work, as well as factors that might affect these efforts [19].

## Overview of the research project

The main aim of the project is to develop and evaluate two sets of digital learning resources to enable young people (secondary school students) to make informed personal choices about their health and to participate as scientifically literate citizens in informed debate about health policies. The project will be implemented in three East African countries: Uganda, Kenya and Rwanda. The overall structure of East Africa's secondary education system is divided into two levels. In the ordinary level (O-level) students achieve a general certificate of education after four years in all East African countries except Rwanda, where it is achieved in three years. The advanced level (A-Level), or upper secondary school, consists of two years for all East African countries except Rwanda, where it is three years. O-level students are generally between 13 and 16 years old, and A-level students are generally between 16 and 18 years old. Secondary schools in East Africa can be public, private, or private with government subsidies. The main language of instruction is English in all three participating East African countries. In this project we will develop English learning resources for O-level students (lower secondary school) in both government and private schools in Uganda, Kenya and Rwanda. The entire project will consist of the following studies, with separate protocols, refer to (Fig 1).

## Objectives of the evaluation of stakeholder engagement

The primary objectives of this evaluation are:

- To ensure that stakeholders are effectively and appropriately engaged in the design, evaluation and dissemination of the learning resources

- To evaluate the extent to which stakeholders were successfully engaged

  Overview of the stakeholder holder engagement process, refer to (Fig 2)

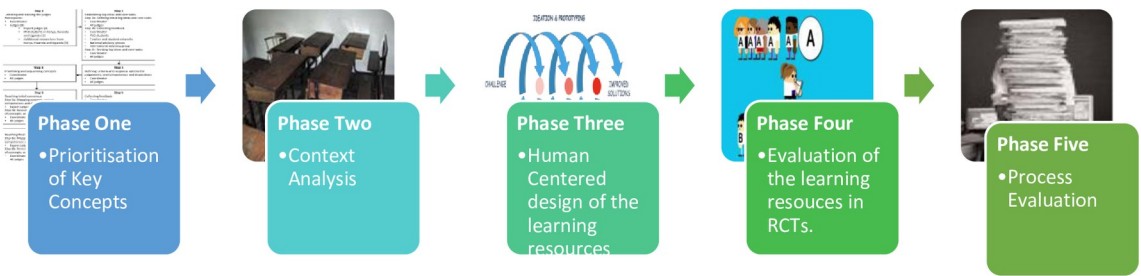

**Fig 1. Overview of the research project.**

## Methods

### Identification and recruitment of stakeholders

The key stakeholders for this project are secondary school teachers and students, i.e. the target users for the IHC secondary school resources. Other stakeholders include school administrators (head teachers and directors), policymakers, curriculum developers, parents, teacher trainers, educational researchers, researchers with expertise in evidence-based medicine or clinical epidemiology, and health professionals. We will engage teachers and students in planning and implementing the project *through teacher and student networks* in each country. We will engage other key stakeholders through a *national advisory panel* in each country, including policymakers, curriculum development officers, school heads, and parent and civil society representatives. In addition, we will engage researchers with expertise in education, health, research methods, design, information and communication technology (ICT) and science communication in an *international advisory panel*. We also will engage colleagues with an interest in adapting, developing, evaluating and implementing IHC resources in other countries through the *IHC network* [21].

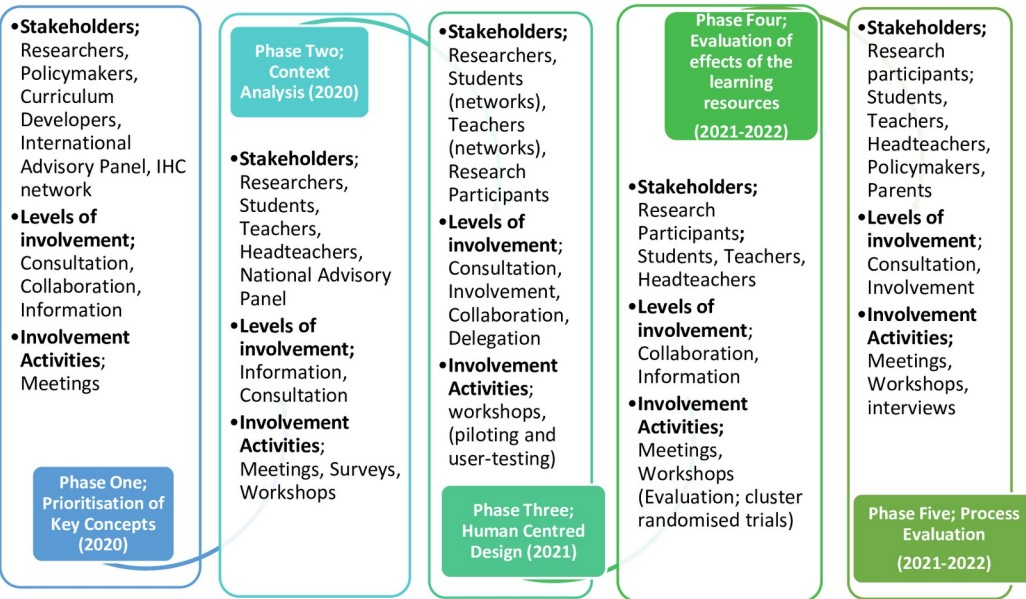

**Fig 2. Overview of the stakeholder engagement process.**

**Prioritisation of key concepts for secondary school learning resources.** We will start with prioritising the IHC Key Concepts [22] that secondary school students need to understand and apply to assess health claims and make informed health choices. Together with teacher and student networks and advisory panels in each country, we will determine which concepts should be taught to lower secondary school students.

**Context analysis.** Under this phase, we will explore issues which can impact use of the new resources, through document analysis and interviews with key informants such as (policy makers, curriculum developers, teacher trainers and parents). The issues to be explored will include:

- exploring the demand for learning resources for teaching critical thinking about health

- mapping where teaching critical thinking about health best fits in the curriculum

- identifying and examining relevant resources already in use

- exploring conditions for introducing new digital learning resources

- identifying opportunities and challenges for developing digital learning resources.

**Human-centred design of the learning resources.** During the development phase, we will employ a user-centred approach to designing the learning resources [23]. This is characterised by iterative cycles of idea generation, prototyping, testing, analysis and revision. The aim is to design resources that teachers and students will experience positively. We will collaborate with a small group of teachers, students and other stakeholders through brainstorming and prototyping workshops [24], school visits, and piloting and user-testing prototypes of the resources. Other stakeholders, including members of national advisory panels, will be informed of progress, and we will invite their feedback and input into the design of the resources.

**Evaluation of the effects of the learning resources.** For the evaluation phase, we will use two-arm, cluster-randomised trials to test the effects of the resources on the students' ability to understand and apply the IHC Key Concepts included in the learning resources. We will select representative samples of 90 to 100 rural and urban schools. The primary outcome measures will be validated tests using multiple-choice questions to measure students' ability to understand and apply the IHC Key Concepts. We may also measure hypothetical decisions, intended behaviours (with hypothetical scenarios), self-assessed ability, attitudes, and actual decisions. Outcomes will be measured at the end of the term in which the learning resources are used and again after one year.

**Process evaluation.** For the final phase of the project, we will conduct process evaluations alongside the trials. The main objectives of the process evaluations will be to: explore the results of the trials; identify factors that may affect the implementation, impact and scaling up of the resources, assuming they are found to be effective; and to identify potential adverse and beneficial effects of the intervention which are not included as outcomes in the trials. We will use structured classroom observations at some of the intervention schools and interviews and focus group discussions at purposively selected schools in each country. We will interview teachers, students, policymakers, and parents. We will use a framework analysis approach to guide data collection and analysis.

**Teacher and student networks.** Building on our experience with a network of primary school teachers in Uganda [3], we will establish a network of secondary school teachers at the start of the project in each of the three countries, Kenya, Rwanda and Uganda. We will also establish networks of students in each country. We will work together with the networks to design, evaluate, and disseminate the resources. The degree of engagement with these groups

will vary from informing them to delegating some decisions to them. Some specific examples are:

- *Informing* them at the start of the project about the project's aims, the IHC Key Concepts, and development and evaluation plans

- *Delegating* to them the final decision about criteria for successful engagement in the project (which input will feed into the content and structure of the interview guides (questionnaires) to be developed for the different stakeholder groups). Any resulting questionnaires will be validated before being used.

- *Consulting* them, i.e. getting their input through interviews and focus groups, regarding the demand for learning resources for teaching students to think critically about health claims and choices; where teaching these skills best fits in the curriculum; and what the conditions are for introducing this into schools, including the availability of time and resources, who the decision-makers are, and what influences their decisions (as described in the protocol for a context analysis)

- *Collaborating* and *consulting* with them, seeking their input into prioritising and sequencing the IHC Key Concepts in workshops (as described in the protocol for prioritisation and sequencing of concepts)

- *Involving* them in generating resource ideas and prototypes together with us in brainstorming and prototyping workshops

- *Consulting* with them about how best to ensure that, if the learning resources achieve their objectives, their use will be scaled up and sustained

- *Consulting* with them and getting their input into choice of success criteria for evaluating the resources

- *Consulting* with them about how best to communicate the findings of our research to teachers, students, and parents

Using a multi-stage stratified sampling method, together with the respective relevant educational authorities, each country team will identify regions that are representative of the country with respect to the key characteristics, then compile a list of lower secondary schools in those districts in the region in close proximity to the study sites (to minimise travel time and expenses). In the second phase, we will randomly sample schools proportionately from the selected districts, stratifying by school setting (rural, semi-urban, or urban) and further by school ownership (public versus private) to ensure a representative sample of secondary schools with varied characteristics that might modify the effects of the learning resources that we will develop. Such factors might include; the type of ICT available in schools and class sizes. We will select a maximum of 25 schools at random per country using online software (www.sealedenvelope.com).

As with our previous study, schools involved in piloting resources will be ineligible for participation in the randomised trial.

For example, the network of primary school teachers in Uganda was selected from a list of schools in four districts near Kampala. We used stratified sampling to select the schools. Government-funded and private schools were identified, and schools were further divided into rural, semi-urban and urban schools. A sample of schools was then randomly selected from the schools in each stratum.

We will recruit between 20 and 25 people for each country student and teacher networks, with a goal of having groups that are sufficiently large to include diverse samples of teachers

and students, while not being so large that it is inefficient and difficult for members of the network to actively engage in discussions and other activities, such as brainstorming workshops. We will therefore restrict the number of participants to not more than 25 per network which is considered a manageable number that would be able to actively participate in the engagement processes [25] In collaboration with the education authority we will send an invitation letter to each school, introducing the project and requesting to meet with the head teacher. When we meet with the head teachers, we will introduce the project and ask them to suggest one or more teachers to join the network. We will suggest that this should be someone who would be interested, take the time to participate, who has teaching experience, and who would be likely to contribute. We will note the gender, level of education, years of experience, and subject areas of the teachers who are suggested. We will use this information to identify a diverse sample of teachers in relation to these criteria. We anticipate that the teachers will primarily be science teachers.

We will send formal letters to the selected teachers, inviting them to be members of the network. The letters will include: a brief introduction to the project; an overview of what we expect of members of the network, what they can expect to gain, and what they can expect of us; timing of the first meeting; and an email address and telephone number for someone whom they can contact for more information.

We will ask each teacher on the three country teachers' networks to suggest one or more students to join their respective student networks. We will suggest that this should be someone who would be interested, be able to participate without adversely affecting the student's schoolwork, and who would be likely to contribute. We will also suggest that the selected students should be a mixture of top, average and low academic performers. We will note the gender and age of the students suggested, again, to ensure a diverse sample.

We will send formal letters to the students and their parents, inviting the students to be members of the network. These letters will include information similar to that outlined in the letters to teachers, consent forms for the parents to sign, and assent forms for the students to sign.

**National advisory panels.** The national advisory panels will include representatives from important groups of stakeholders in each of the three participating countries. The research team in each country will contact key groups, such as the national curriculum committees, by phone or email and arrange face-to-face meetings to inform them about the projects' aims and objectives. Those groups will be asked to identify suitable members for the national advisory panels and to identify other key stakeholders that we should contact.

We will aim to identify 15 individuals in each country with a range of experience, including: policymakers at the ministerial and regional or district levels, members of the national curriculum committees, school directors, head teachers, leaders of teacher unions, and representatives of parents' groups and civil society. We will include representatives of groups of stakeholders where the group is likely to have an interest in being engaged, there are risks not engaging the group, and the group is likely to have important input into designing, evaluating, or disseminating the learning resources.

Criteria for selecting individuals to represent the different groups of stakeholders include selection by their organisation or being influential within their organisation, being interested, willing to take the time to participate, and being likely to contribute. We will consult informal contacts, the teacher networks, and initial invitees (snowballing). We will purposively select members to ensure a diverse range of experience and expertise. Where we are unable to recruit representatives of a group, we will report this, in the evaluation of stakeholder engagement and discuss the implications.

We will collaborate with the national advisory panels; i.e. seek their advice and incorporate it as far as possible. This includes advice on:

- The criteria for success of their engagement in the project (allowing them to decide on the final criteria to be used)

- Protocols for each element of the project

- How best to connect the IHC secondary school resources to the curriculum.

- How best to ensure that use of the resources will be scaled up and sustained, given they are effective

- Success criteria for the evaluation of the resources

- How best to communicate the findings of our research to key stakeholder groups

**International advisory panel.** The international advisory panel will include about 30 people with experience and expertise in education and education policy, relevant areas of research (e.g. education, health literacy, science communication, evidence-based practice, critical appraisal and critical thinking), design, ICT, and learning games, as well as representatives of key international organisations. We will identify potential members through personal contacts, relevant publications, and snowballing. We will aim to include people from a wide range of countries, including low-, middle-, and high-income countries.

We will contact potential members by email, informing them about the project, what is expected of members of the international advisory panel, and what they can expect from us. The international advisory panel will be engaged through consultation. We will seek their input and feedback on protocols, prioritisation of Key Concepts, the design and evaluation of the learning resources, and dissemination. A key role of this group will be to help us ensure that, with a minimum of modification, the learning resources and our research findings have the potential to be used in countries other than the three countries where they will be initially developed and evaluated.

**IHC network.** In addition to consulting the international advisory panel, we will consult the IHC network [21]. The IHC network is an informal collaboration of people in over 20 countries who are developing, evaluating or contextualising IHC primary school resources. It is an international, multidisciplinary group with experience in research methods, health services research, medicine, public health, epidemiology, design, education, communication and journalism. It includes contributors in Arab-speaking countries, Australia, Basque Country, Brazil, Chile, China, Croatia, French-speaking countries, German-speaking countries, Iran, Israel, Ireland, Italy, Kenya, Mexico, Norway, Poland, Rwanda, South Africa, Spain, Uganda, United Kingdom, and United States. In addition to the International advisory group, the IHC network will help ensure that the learning resources will have a broader appeal in the different contexts but, in particular, in their own countries.

## Engagement of research participants

In addition to engaging stakeholders as described above, we will engage research participants in the design and evaluation of the learning resources. Teacher and student participants who are not necessarily members of the teacher and student networks will be *consulted* and *involved* in designing the resources through the user testing and piloting.

Participants in the trial will not be engaged in the design of the resources or planning the research. They will be *informed* about the resources and the trial during the trial recruitment process. After the trial is completed they will be informed about the results.

Participants in the process evaluation will be *consulted* and *involved* in interpreting the results of the trial identifying potential adverse and beneficial effects of the intervention. If desired effects are achieved this will involve identifying factors that may affect the implementation, impact and scaling up of the intervention.

## Success criteria

Together with the teacher and student networks and the advisory groups, we will establish measurable success criteria that reflect the objectives of engaging stakeholders at the start of the project and evaluate the extent to which those criteria were met at the end of the project [15]. We will consider criteria related to whether:

- The stakeholders were informed and engaged to an appropriate extent

- The approaches (of informing and engaging) that were used were appropriate and worked as expected

- The level of involvement was appropriate

- The input was appropriate and whether it was used appropriately

- The intended outputs were delivered and appropriate

- The intended outcomes were achieved

- The efforts were worthwhile relative to what was achieved

- The appropriateness of group make-up (were important voices missing/not represented)

[the words 'appropriate' and 'worthwhile' beg questions about how these words will be defined and how judgements will be made]

The final decision about the success criteria will be delegated to the networks and advisory groups.

We will also discuss with them how we plan to engage stakeholders and their perceptions to judge whether our plans are likely to achieve success by the agreed criteria. The final decision about these plans and any changes that are made during the project will be made collaboratively with the networks and advisory panels.

## Data collection

The evaluation will include a basic description of what was done, including the objectives, and how stakeholders were identified and engaged. This description will include our plan at the start of the project, modifications made following discussions with the networks and advisory groups, and any deviations or modifications made during the project, including the reasons for those.

We will use questionnaires and semi-structured interviews to collect feedback from the stakeholders and from the research team. The questionnaires will reflect the agreed upon criteria for success. Separate questionnaires will be prepared for and completed by the teacher networks, the student networks, the advisory panels, and participants in the research. Decisions about what information to collect from different stakeholders and participants and whom to interview will be made collaboratively with the networks and advisory panels. The interviews may be conducted in English or a local language (Luganda, Swahili or Kinyarwanda), based on the preference of the person being interviewed. Interviews will be recorded, transcribed, and, if necessary, translated to English. Meetings with the networks and advisory groups will be in English. At least two members of the research team will participate in discussions, one of

whom will take notes. The notes will be shared with and approved by the relevant network or advisory panel.

## Analysis

We will use an interpretative description approach to collecting and analysing the data [26]. This approach borrows from grounded theory, naturalistic inquiry, and ethnography. It differs from these approaches in that the investigators are looking for findings with practical applications. Various verification strategies, such as concurrent data collection and analysis, constant comparative analysis and iterative analysis, serve to locate the findings within the framework of the existing body of knowledge. The object of the exercise is a coherent conceptual description that can inform effective strategies for engaging stakeholders in this and other research.

We will enter findings into a spreadsheet after each engagement process. Atleast two of the researchers will independently code each finding based on the importance of the finding (Table 2) and its implications for changes to the engagement processes.

We will also present and discuss findings and interpretation of the findings with the networks and advisory panels iteratively. We will record and report any discrepancies between how findings are interpreted by the research team and other stakeholders, any disagreements that have not been resolved by discussion within each group, and differences in interpretation among groups or countries.

## Interpretation and reporting of findings

The networks and advisory groups will be engaged in interpreting and confirming interpretations of the findings. They will be invited to comment on reports and be co-authors.

We will seek approval of the networks and advisory groups of a final report of the evaluation of stakeholder engagement in this project.

When reporting the results, we will use the GRIPP2 Checklist for reporting patient and public involvement in health and social care research [27].

## Ethical considerations

Ethics approval has been sought from local institutional review boards and national research governing councils in each country. We have obtained ethics approval from the Rwanda National Ethics Committee; approval number 916/RNEC/2010- for the Rwandan study site,

**Table 2. Coding of the importance of observations and feedback.**

| Code | Description |
|---|---|
| Very important negative finding ("show stopper") | A problem that we should address for the stakeholder engagement processes to be appropriate and effective |
| Important negative finding | A problem that we should probably address for part of the stakeholder engagement processes to be effective |
| Negative finding | A problem that we can easily address and probably will not prevent the stakeholder engagement processes from being effective |
| Very important positive finding | Praise that probably should inspire changes |
| Important positive finding | Praise that maybe should inspire changes |
| Positive finding | Praise that probably should not inspire changes |
| Very important constructive finding | A suggestion that probably should inspire changes |
| Important constructive finding | A suggestion that maybe should inspire changes |
| Constructive finding | A suggestion that probably should not inspire changes |

and from Masinde University of Science and Technology Institutional Ethics Review Committee and the Kenyan National Commission for Science, Technology and Innovation; approval number NACOSTI/P119/1986 for the Kenyan study site. We submitted our application to the Makerere University College of Health Sciences IRB whose comments and suggestions have been addressed and are currently awaiting final approval from the Uganda National Council of Science and Technology for the Ugandan study site. Data collection will commence once the study has been approved by all three participating sites.

We will obtain informed consent for all interviews. Data from questionnaires, interviews, photos, videos and notes from discussions will be anonymised.

Teachers, students and policymakers who are invited to participate in any of the networks will be informed of the purpose of their participation before written permission is sought. The students who participate on the networks and user-testing will be given information about the project to take to their parents, where written permission will be obtained both from the parents in form of consent and from the students in form of assent.

Consent for the students to participate in the pilot will be given by the headteachers and teachers and students and their parents will have the same right to refuse participation in piloting the resources as they do for any other learning resources used in schools. Although this will be limited, teachers, students and policymakers, whenever required to attend meetings off-site of their professional premises (schools, ministry offices etc) will be compensated for their transport expenses.

## Choice of methods, risks and risk management

Although there are strong arguments for engaging stakeholders in research, there is a paucity of research comparing different ways of doing this [9]. By having explicit goals for engaging stakeholders and monitoring the extent to which those goals are being achieved, we will be able to make adjustments as needed throughout the project. Our decision to engage teachers and students through a network in each country is based on our prior experience with a teacher network [3]. We have prioritised engaging teachers and students because they are the intended users of the resources that we will develop.

We will consider and discuss important risks of stakeholder engagement with stakeholders, to plan how we will manage those risks, including potential waste of resources, loss of credibility, conflicts, and adverse effects on the design, evaluation, or dissemination of the learning resources.

## Author Contributions

**Conceptualization:** Andrew David Oxman.

**Funding acquisition:** Andrew David Oxman.

**Investigation:** Allen Nsangi, Andrew David Oxman, Matt Oxman, Sarah E. Rosenbaum, Daniel Semakula, Ronald Ssenyonga, Michael Mugisha, Faith Chelagat, Margaret Kaseje, Leaticia Nyirazinyoye.

**Supervision:** Andrew David Oxman.

**Writing – original draft:** Allen Nsangi.

**Writing – review & editing:** Andrew David Oxman, Matt Oxman, Sarah E. Rosenbaum, Daniel Semakula, Ronald Ssenyonga, Michael Mugisha, Faith Chelagat, Margaret Kaseje, Leaticia Nyirazinyoye, Iain Chalmers, Nelson Kaulukusi Sewankambo.

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
