## [Decision Letter · Decision Letter 0]

2 Jul 2020

PONE-D-20-04939

Protocol for evaluating stakeholder engagement in the design, evaluation, and dissemination of the Informed Health Choices resources for helping secondary school students learn to think critically about health claims and choices

PLOS ONE

Dear Dr. Oxman,

Thank you for submitting your manuscript to PLOS ONE. After careful consideration, we feel that it has merit but does not fully meet PLOS ONE’s publication criteria as it currently stands. Therefore, we invite you to submit a revised version of the manuscript that addresses the points raised during the review process.

Please note that both reviewers have very substantial comments and the expected revision is significant both from a conceptual and methodological description level. Should you decide to revise and resubmit the paper, it will be re-reviewed.

We look forward to receiving your revised manuscript.

Kind regards,

Sara Rubinelli

Academic Editor

PLOS ONE

"We will seek ethics approval from local institutional review boards and national research governing councils in each country. We will obtain informed consent for all interviews. Data from questionnaires, interviews, photos, videos and notes from discussions will be anonymised.

Teachers, students and policymakers who are invited to participate in one of the six networks will be informed of the purpose of their participation before written permission is sought. The students who participate on the networks and user-testing will be given information about the project to take to their parents, where written permission will be obtained both from the parents in form of consent and from the students in form of assent.

Consent for the students to participate in the pilot will be given by the headteachers and teachers and students and their parents will have the same right to refuse participation in piloting the resources as they do for any other learning resources used in schools. Although this will be limited, teachers, students and policymakers, whenever required to attend meetings off-site of their professional premises (schools, ministry offices etc) will be compensated for their transport expenses.".

5. Please ensure that you refer to Figure 1 and 2 in your text as, if accepted, production will need this reference to link the reader to each figure.

6. We note that Figure 1 includes an image of participants in the study.

As per the PLOS ONE policy (http://journals.plos.org/plosone/s/submission-guidelines#loc-human-subjects-research) on papers that include identifying, or potentially identifying, information, the individual(s) or parent(s)/guardian(s) must be informed of the terms of the PLOS open-access (CC-BY) license and provide specific permission for publication of these details under the terms of this license.

Please download the Consent Form for Publication in a PLOS Journal (http://journals.plos.org/plosone/s/file?id=8ce6/plos-consent-form-english.pdf). The signed consent form should not be submitted with the manuscript, but should be securely filed in the individual's case notes.

Please amend the methods section and ethics statement of the manuscript to explicitly state that the patient/participant has provided consent for publication: “The individual in this manuscript has given written informed consent (as outlined in PLOS consent form) to publish these case details”.

Reviewers' comments:

Reviewer's Responses to Questions

**Comments to the Author**

1. Does the manuscript provide a valid rationale for the proposed study, with clearly identified and justified research questions?

Reviewer #1: Partly

Reviewer #2: Yes

2. Is the protocol technically sound and planned in a manner that will lead to a meaningful outcome and allow testing the stated hypotheses?

Reviewer #1: Partly

Reviewer #2: Partly

3. Is the methodology feasible and described in sufficient detail to allow the work to be replicable?

Reviewer #1: No

Reviewer #2: Yes

4. Have the authors described where all data underlying the findings will be made available when the study is complete?

Reviewer #1: No

Reviewer #2: Yes

5. Is the manuscript presented in an intelligible fashion and written in standard English?

Reviewer #1: Yes

Reviewer #2: Yes

6. Review Comments to the Author

You may also provide optional suggestions and comments to authors that they might find helpful in planning their study.

Reviewer #1: This paper addresses a very interesting and timely topic, and this is not just for the project aims but also for the overall context in which it develops. However, the manuscript is often too generic when presenting future actions, which is a bit odd for a protocol paper. I do not think the manuscript in its present form it is suitable for publication. Below my main concerns.

• In the Background section the authors point to differences in the concepts of public involvement and engagement, both in terms of definitions and approaches and provide their working definition. They also report on models and framework of engagement, but the aim of the paper is not explicit. I believe this digression, even though interesting, could be definitely condensed, as this is not a perspective or position paper but a protocol.

• The overview of the project is also very detailed and appreciated, however it should be integrated in the methods to my mind. Again, it seems odd to me to read the description of the objectives at the end of page 5.

• Methods section is the most problematic part of the paper. The authors state their objective clearly, the multi-stage stratified sampling method is clearly exemplified. However, I feel that the objectives of each phase, the exact process of involvement, and the methods used for collecting and analyzing data should be more precisely presented. This section is at present too discursive.

• Going back to the objectives of the study: they seem to be in contradiction. On the one hand the aim is to ensure that stakeholders are appropriately engaged, and on the other is to evaluate the extent to which they were successfully engaged. It may be that the first objective follows a top-down reasoning on engagement and design the process (and here the focus would be on how this has been informed and which are the motivations behind), and the second one is more of a bottom-up (understanding if it actually worked). However, I believe that the two may be difficult to be represented in the same study, and it is actually maybe the reason behind what now seem to me mix of input from the literature and good ideas for a project planning (but not yet a protocol).

Reviewer #2: Unfortunately, the title is too long. Please consider reducing it’s length considerably.

This manuscript is a registers report protocol. The authors of this manuscript should know that while this reviewer is an experienced reviewer, this represents the first time that I’ve reviewed this type of manuscript.

Health claims and choices, it’s not quite clear what that means.

Page 2, Abstract. It’s not clear how long this project is projected to take. This seems like an important piece of information to include in the Abstract.

Page 3, Background. The reader doesn’t know what the Informed Health Choices Project is. You are talking about thinking critically about health claims and choices, but you do not give us any information as to what exactly that is that was done in these meetings and what it is that you discovered.

Page 8, Figure 2. This is the core of the stakeholder engagement process. As indicated above, it is important to understand the timescale of these phases.

Page 8. It would seem that a snowball sampling technique might be useful to occupy enough spaces in each strata, but it’s just not clear how the researchers will know when their sample is representative.

Page 12. Success Criteria. It’s not clear how you are going to measure any of these criteria.

Page 12, Analysis. Yes, this is an interpretative description approach, but it’s unclear how that qualitative data will be analyzed. Researchers should have some idea of what analysis techniques you are planned to use for this excellent research approach for applied approach for understand human health. It’s inductive, but that doesn’t mean it’s absent from a statistical analytic method of analysis, if that’s what you are planning on doing.

The research claims it will use a multi-stage stratified sampling method, This method is best when subpopulations are expected to yield response differences that are significant. What is the motivation for this expected differences? Another related question is what analytic method are you going to use to analyze the data? How are you going to account for the validity (precision) and reliability (consistency) of the responses that you do get? Furthermore, it’s not clear what your sample size is going to be and how you are going to determine what the proper sample size is for your research?

7. PLOS authors have the option to publish the peer review history of their article (what does this mean?). If published, this will include your full peer review and any attached files.

Reviewer #1: No

Reviewer #2: **Yes: **John Lawrence Dennis

---

## [Author Response · Author response to Decision Letter 0]

10 Aug 2020

Comments from the reviewers:

Reviewer 1:

This paper addresses a very interesting and timely topic, and this is not just for the project aims but also for the overall context in which it develops. However, the manuscript is often too generic when presenting future actions, which is a bit odd for a protocol paper. I do not think the manuscript in its present form it is suitable for publication.

Response: Thank you very much for the constructive feedback, most of which has been taken on board to enrich the manuscript.

1. In the Background section the authors point to differences in the concepts of public involvement and engagement, both in terms of definitions and approaches and provide their working definition. They also report on models and framework of engagement, but the aim of the paper is not explicit. I believe this digression, even though interesting, could be definitely condensed, as this is not a perspective or position paper but a protocol.

Response: This has been addressed on page 3.

2. The overview of the project is also very detailed and appreciated, however it should be integrated in the methods to my mind. Again, it seems odd to me to read the description of the objectives at the end of page 5.

Response: This has been addressed on page on page 7 and 8.

3. Methods section is the most problematic part of the paper. The authors state their objective clearly, the multi-stage stratified sampling method is clearly exemplified. However, I feel that the objectives of each phase, the exact process of involvement, and the methods used for collecting and analyzing data should be more precisely presented. This section is at present too discursive.

Response: This has been addressed on page on page 8, 9 and 13.

4. Going back to the objectives of the study: they seem to be in contradiction. On the one hand the aim is to ensure that stakeholders are appropriately engaged, and on the other is to evaluate the extent to which they were successfully engaged. It may be that the first objective follows a top-down reasoning on engagement and design the process (and here the focus would be on how this has been informed and which are the motivations behind), and the second one is more of a bottom-up (understanding if it actually worked). However, I believe that the two may be difficult to be represented in the same study, and it is actually maybe the reason behind what now seem to me mix of input from the literature and good ideas for a project planning (but not yet a protocol).

Response: It is unclear why the reviewer finds these two objectives to be in contradiction or why the reviewer perceives the first as “top down” and the second as “bottom up”. It is correct that we, the researchers, have determined that an objective is to ensure that stakeholders are appropriately engaged. However, we have not made a “top down” decision about what is appropriate. As stated under “Success Criteria”: “The final decision about the success criteria will be delegated to the networks and advisory groups.” Having determined success criteria together with stakeholders, we will then evaluate the extent to which those criteria were met. These two objectives – trying to ensure that stakeholders are effectively and appropriately engaged (together with stakeholders) and evaluating the extent to which we were successful in doing this are logically linked and supported by the literature that we cite.

The argument that this is “not yet a protocol” suggests that we should wait until after the success criteria have been determined and questionnaires have been designed before publishing our protocol. We agree this would be an option. However, it is unclear why the reviewer would prefer that we describe how we determined the success criteria and the design of the questionnaires retrospectively (without a protocol).

Reviewer 2:

Response: Thank you very much for the constructive feedback, most of which has been taken on board to enrich the manuscript.

1. Unfortunately, the title is too long. Please consider reducing its length considerably.

Response: This has been addressed on page 1.

2. This manuscript is a registers report protocol. The authors of this manuscript should know that while this reviewer is an experienced reviewer, this represents the first time that I’ve reviewed this type of manuscript.

Health claims and choices, it’s not quite clear what that means.

Response: This has been addressed on page 3.

3. Page 2, Abstract. It’s not clear how long this project is projected to take. This seems like an important piece of information to include in the Abstract.

Response: This has been addressed on page 2.

4. Page 3, Background. The reader doesn’t know what the Informed Health Choices Project is. You are talking about thinking critically about health claims and choices, but you do not give us any information as to what exactly that is that was done in these meetings and what it is that you discovered.

Response: The detail about the Informed Health Choices project has been availed on page 3, but with this being a protocol, we have not yet embarked on the study and would not be in a position to share any findings at this point.

5. Page 8, Figure 2. This is the core of the stakeholder engagement process. As indicated above, it is important to understand the timescale of these phases.

Response: This has been addressed on page 7.

6. Page 8. It would seem that a snowball sampling technique might be useful to occupy enough spaces in each strata, but it’s just not clear how the researchers will know when their sample is representative.

Response: This has been clarified on page 9.

7. Page 12. Success Criteria. It’s not clear how you are going to measure any of these criteria.

Response: We agree that it is not clear how we will measure the success criteria at this time, since the success criteria have not yet been determined. We have, however, described how the success criteria will be determined (with the final decision delegated to stakeholders), provided a framework for identifying success criteria (under Success Criteria), and described the use of “questionnaires and semi-structured interviews to collect feedback from the stakeholders and from the research team.” “The questionnaires will reflect the agreed upon criteria for success.” “Decisions about what information to collect from different stakeholders and participants and whom to interview will be made collaboratively with the networks and advisory groups.” As noted above, we could wait to publish a protocol until after the success criteria are determined and the questionnaires and interview guides are designed. However, determination of success criteria is a result and we believe that the advantages of describing a priori how we will determine the success criteria, what information to collect and how outweigh any perceived advantages of waiting until after this is done to publish a protocol.

8. Page 12, Analysis. Yes, this is an interpretative description approach, but it’s unclear how that qualitative data will be analyzed. Researchers should have some idea of what analysis techniques you are planned to use for this excellent research approach for applied approach for understand human health. It’s inductive, but that doesn’t mean it’s absent from a statistical analytic method of analysis, if that’s what you are planning on doing.

Response: This has been addressed on page 13.

9. The research claims it will use a multi-stage stratified sampling method, This method is best when subpopulations are expected to yield response differences that are significant. What is the motivation for this expected differences? Another related question is what analytic method are you going to use to analyze the data? How are you going to account for the validity (precision) and reliability (consistency) of the responses that you do get? Furthermore, it’s not clear what your sample size is going to be and how you are going to determine what the proper sample size is for your research?

Response: This has clarified on page 9.

---

## [Decision Letter · Decision Letter 1]

17 Sep 2020

Protocol for assessing stakeholder engagement in the development and evaluation of the Informed Health Choices resources teaching secondary school students to think critically about health claims and choices

PONE-D-20-04939R1

Dear Dr. Oxman,

We’re pleased to inform you that your manuscript has been judged scientifically suitable for publication and will be formally accepted for publication once it meets all outstanding technical requirements.

Kind regards,

Sara Rubinelli

Academic Editor

PLOS ONE

Additional Editor Comments (optional):

Reviewers' comments:

Reviewer's Responses to Questions

**Comments to the Author**

1. Does the manuscript provide a valid rationale for the proposed study, with clearly identified and justified research questions?

Reviewer #1: Yes

Reviewer #2: Yes

2. Is the protocol technically sound and planned in a manner that will lead to a meaningful outcome and allow testing the stated hypotheses?

Reviewer #1: Yes

Reviewer #2: Yes

3. Is the methodology feasible and described in sufficient detail to allow the work to be replicable?

Reviewer #1: Yes

Reviewer #2: Yes

4. Have the authors described where all data underlying the findings will be made available when the study is complete?

Reviewer #1: No

Reviewer #2: Yes

5. Is the manuscript presented in an intelligible fashion and written in standard English?

Reviewer #1: Yes

Reviewer #2: Yes

6. Review Comments to the Author

You may also provide optional suggestions and comments to authors that they might find helpful in planning their study.

Reviewer #1: The authors have reviewed their manuscript by taking into account and addressing appropriately almost all reviewers' comments. I also find acceptable the response to my fourth comment.

Reviewer #2: Thank you for following all of the recommendations that have been provided. This reviewer is grateful for the authors efforts.

7. PLOS authors have the option to publish the peer review history of their article (what does this mean?). If published, this will include your full peer review and any attached files.

Reviewer #1: No

Reviewer #2: **Yes: **John Lawrence Dennis

---

## [Editor Report · Acceptance letter]

21 Sep 2020

PONE-D-20-04939R1 

Protocol for assessing stakeholder engagement in the development and evaluation of the Informed Health Choices resources teaching secondary school students to think critically about health claims and choices 

Dear Dr. Oxman:

I'm pleased to inform you that your manuscript has been deemed suitable for publication in PLOS ONE. Congratulations! Your manuscript is now with our production department. 

Kind regards, 

on behalf of

Dr. Sara Rubinelli 

Academic Editor

PLOS ONE